# Matryoshka Query Transformer for Large Vision-Language Models

**Wenbo Hu   Zi-Yi Dou   Liunian Harold Li   Amita Kamath**
**Nanyun Peng   Kai-Wei Chang**
University of California, Los Angeles
{whu,zdou,liunian.harold.li,kamatha}@cs.ucla.edu

https://github.com/MQT-LLaVA

## Abstract

Large Vision-Language Models (LVLMs) typically encode an image into a fixed number of visual tokens (e.g., 576) and process these tokens with a language model. Despite their strong performance, LVLMs face challenges in adapting to varying computational constraints. This raises the question: can we achieve flexibility in the number of visual tokens to suit different tasks and computational resources? We answer this with an emphatic *yes*. Inspired by Matryoshka Representation Learning, we introduce the Matryoshka Query Transformer (MQT), capable of encoding an image into $m$ visual tokens during inference, where $m$ can be *any* number up to a predefined maximum. This is achieved by employing a query transformer with $M$ latent query tokens to compress the visual embeddings. During each training step, we randomly select $m \leq M$ latent query tokens and train the model using only these first $m$ tokens, discarding the rest. Combining MQT with LLaVA, we train a single model once, and flexibly and drastically reduce the number of inference-time visual tokens while maintaining similar or better performance compared to training independent models for each number of tokens. Our model, MQT-LLAVA, matches LLaVA-1.5 performance across 11 benchmarks using a maximum of 256 tokens instead of LLaVA's fixed 576. Reducing to 16 tokens (8x less TFLOPs) only sacrifices the performance by 2.4 points on MMBench. On certain tasks such as ScienceQA and MMMU, we can even go down to only 2 visual tokens with performance drops of just 3% and 6% each. Our exploration of the trade-off between the accuracy and computational cost brought about by the number of visual tokens facilitates future research to achieve the best of both worlds.

## 1   Introduction

Recent work in Large Vision-Language Models (LVLMs) (OpenAI, 2023; Liu et al., 2023b; Bai et al., 2023) has shown remarkable performance across a broad range of vision-language tasks (Huang et al., 2023; Chen et al., 2023; Cai et al., 2024a; Li et al., 2023b). These LVLMs typically consist of a vision encoder to embed images into grid features, which are fed into a Large Language Model (LLM) (Touvron et al., 2023; Chiang et al., 2023) for processing and reasoning alongside a text input.

A key research question is how to transform these raw visual embeddings into the visual tokens that are fed into the LLM. Prior work either directly projects the grid features with a multi-layer perceptron (MLP) (Liu et al., 2023b) or compresses the grid features into fewer tokens with a query transformer or resampler (Li et al., 2023a; Dai et al., 2023; Bai et al., 2023; Ye et al., 2023; Alayrac et al., 2022). However, these models all need to pre-determine how many tokens an image is worth, and set a fixed number for all images. Finding a *flexible number* that adaptively strikes a balance

38th Conference on Neural Information Processing Systems (NeurIPS 2024).

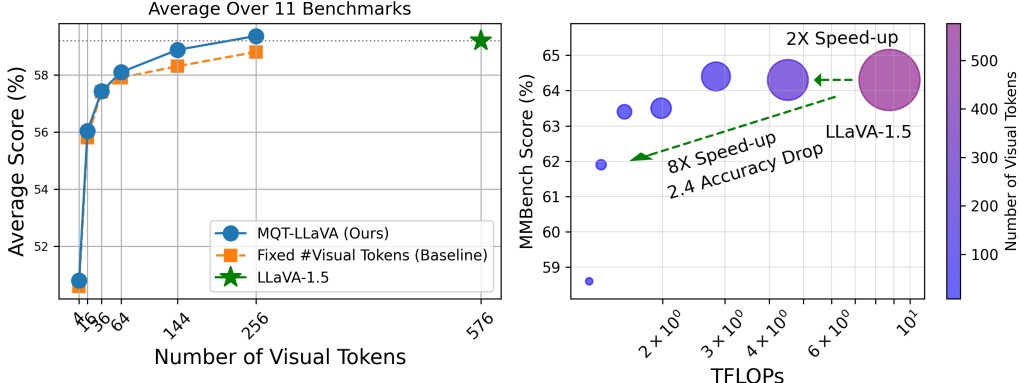

Figure 1: Our model, MQT-LLAVA, matches LLaVA-1.5 performance on 11 benchmarks using only 256 visual tokens instead of 576. We achieve a 2x speed-up with 256 tokens and 8X speed-up in TFLOPs using 16 tokens with only a 2.4 performance drop compared to LLaVA-1.5 on MMBench.

between efficiency and performance is difficult. More visual tokens encode more information, but come at a higher inference cost, as the complexity of the transformers used in these LVLMs scales quadratically with the number of input tokens. Additionally, not all applications require or allow the same token budget: some applications have limited computational resources, necessitating a lower token budget to ensure real-time processing. In practice, most best-performing LVLMs choose a fixed, large number of visual tokens per image (e.g., 576 for LLaVA-1.5) without the ability to adaptively adjust the visual token allocation at deployment time.

In this work, inspired by Matryoshka Representation Learning (MRL) (Kusupati et al., 2022; Kudugunta et al., 2023), we introduce Matryoshka Query Transformer (MQT), a simple way to train a single LVLM that supports adaptively changing the number of visual tokens at inference time. We use a query transformer (Li et al., 2022; Alayrac et al., 2022) with $M$ latent query tokens to transform grid features into visual tokens. Crucially, during each training step, we train the model using only the first $m$ latent query tokens while dropping the rest, where $m$ is randomly selected within the range of $M$. With such a tail-token dropping strategy, the query tokens form a Matryoshka structure. Intuitively, the significance of each token correlates with its placement within this nested structure. During inference, we have the flexibility to selectively utilize solely the initial $m$ visual tokens.

We combine MQT with LLaVA-1.5: the resulting model, MQT-LLAVA, is able to match LLaVA-1.5 performance across 11 benchmarks using only a maximum of 256 tokens, instead of LLaVA's fixed 576. When the maximum number of tokens is dropped drastically to only 2 tokens, MQT-LLAVA performance drops by only 3% on ScienceQA and 6% on MMMU. Finally, we study the performance of 2, 4, 8, 16, 36, 64, 144, and 256 visual tokens during inference across 11 benchmarks, and offer a trade-off in the selection of visual tokens that balances achieving the highest accuracy with minimizing computational costs on different tasks. Interestingly, we find that changing the number of visual tokens impacts different tasks very differently. For instance, tasks involving language-based reasoning and subject-level scientific knowledge can achieve excellent performance with only a few tokens, whereas complex open-ended visual question tasks that involve rich local information details require a larger number of tokens.

In summary, we make the following key contributions:

- We introduce Matryoshka Query Transformer (MQT), which allows for a flexible choice of the number of visual tokens and accommodates varying computational constraints in different tasks.

- Leveraging MQT, we build MQT-LLAVA, a vision-language model that matches the performance of LLaVA-1.5 using less than half the number of visual tokens, and outperforms it in 6 out of 11 benchmarks.

- We further explore the performance and computation trade-offs across 11 tasks and demonstrate that a significant speed-up can be achieved with minimal performance drop by reducing the number of visual tokens (e.g., 8X fewer TFLOPs with 2.4 points drop on MMBench).

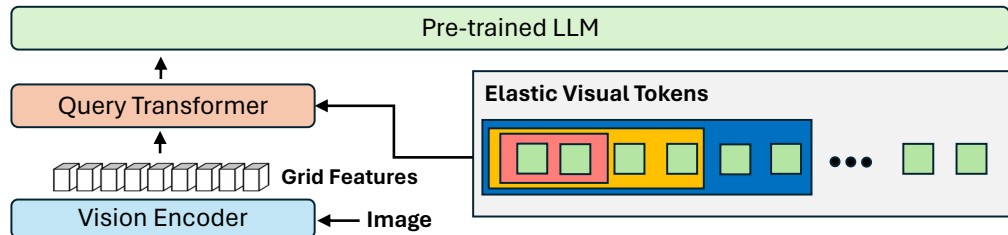

Figure 2: Our model employs a query transformer to encode images as visual tokens. We randomly select the first $m$ tokens during training, and enable flexible choice of *any* $m$ number under $M$ during inference, where $M$ is the maximum number of initialized tokens.

## 2 Matryoshka Query Transformer

**Preliminary: Matryoshka Representation Learning (MRL).** MRL (Kusupati et al., 2022; Kudugunta et al., 2023) involves training models with nested dimensions to learn representations at multiple granularities, enabling adaptive deployment per computational constraints. MRL defines a series of models $f_1, f_2, \ldots, f_M$ with the same input and output space but growing hidden dimensions.

The name "Matryoshka" comes from the fact that the parameters of $f_m$ are contained by $f_{m+1}$. For example, in Kudugunta et al. (2023), $\{f_m\}$ are a series of Transformers with the same depth but different widths. Consider a specific Feed Forward Network (FFN) block in $f_M$ that has $d_M$ neurons in the hidden layer. Then, the FFN block in $f_m$ will contain the first $d_m$ neurons, and $d_1 \leq d_2 \leq \cdots \leq d_M$. MRL then trains these models jointly with the following loss:

$$\sum_m c_m \cdot \mathcal{L}\left(f_m(x);\ y\right), \tag{1}$$

where $\mathcal{L}$ is the loss function and $y$ is the ground truth label. Note that for each training step, MRL performs forward and backward passes for all $M$ models, inducing significant training overhead compared to training one model. After training, MRL can perform inference with any hidden dimension $d_{i \leq M}$, enabling flexible deployment based on specific needs. MRL is our motivation to train LVLMs that can perform inference with a flexibly selected number of visual tokens.

### 2.1 MQT-LLaVA

We first explain how we encode images with a query transformer, then discuss our training paradigm.

**Encoding images with a Query Transformer.** We employ a query transformer-based architecture to extract visual tokens from images following previous work (Li et al., 2022; Bai et al., 2023). Specifically, an input image $x$ is first processed by an image encoder and are then flattened into $H \times W$ grid features $\mathbf{G} = [\mathbf{g}_{11}, \cdots, \mathbf{g}_{1W}, \cdots, \mathbf{g}_{H1}, \cdots, \mathbf{g}_{HW}]$. Then, a query transformer $Q$ is applied to compress the grid features to $M$ visual tokens. Specifically, $Q$ assumes a set of latent *query tokens* $\mathbf{Z} = [\mathbf{z}_1, \ldots, \mathbf{z}_M]$ as input, where $M$ is usually smaller than $H \times W$. The query tokens cross-attend to the grid features and compress the information into the query tokens. The final-layer query tokens become the visual tokens $\mathbf{V}$ that are fed to a large language model together with the input text tokens. I.e., $\mathbf{V} = Q(\mathbf{Z}, \mathbf{G})$. A linear projection layer is added in the end to match the hidden size of the language model.[1]

**Matryoshka Query Transformer.** To enable elastic inference, given the $M$ latent query tokens $\mathbf{Z} = [\mathbf{z_1}, \ldots, \mathbf{z_M}]$, at each training step, we feed only the first $m$ query tokens to the query transformer $Q$. Subsequently, we obtain only $m$ visual tokens from the query transformer. $m$ can be any number equal to or smaller than the maximal token number $M$. In practice, we choose $m$ from a linear set of maximum dimensions, in increments of 2, e.g. $m$ can be any number in $\{2, 4, 6, \ldots, 252, 254, 256\}$

---

[1]Unlike previous work (Bai et al., 2023; Ye et al., 2023) that first applies projection followed by attention, we empirically find that our "attention then projection" architecture performs better (c.f. §4.3).

when $M = 256$. From a training efficiency perspective, our approach uses, on average, half of the visual tokens compared to the original query transformer-based models.

Formally, given an input image with its corresponding text question $q$ and answer $y$, at each training step, we randomly select a $m$ and feed the first $m$ latent tokens $\mathbf{Z}_{1:m}$ and the text question $q$ to the model. We compare the model output and $y$ and minimize

$$c_m \cdot \mathcal{L} \left( \text{LM}(\mathbf{V}, q);\ y \right),\ \text{where } \mathbf{V} = Q(\mathbf{Z}_{1:m}, \mathbf{G}), \tag{2}$$

LM is the language model, $\mathcal{L}$ is the language modeling loss function, and $c_m$ is a constant coefficient to control the weight of different numbers of visual tokens, which is always set to 1 in our setting.

**Discussion.** Here we discuss several interesting properties of MQT. (1) Unlike the original matryoshka representation learning that maintains a nested structure in the parameter space, we specifically target LVLMs and make the visual tokens Matryoshka-like. (2) Despite discarding the tail $M - m$ tokens during each training step, models trained with this token-dropping strategy perform comparably to those trained consistently with all $M$ tokens, as long as we utilize the entire $M$ tokens during inference for both models. (3) Unlike the original MRL, which performs forward and backward passes for all $M$ configurations in each step, we now select just one model configuration per training step, significantly cutting training costs. (4) Our cost reduction enables training across a broader spectrum of $m$ values, facilitating the training of models with a more diverse range of choices compared to the original MRL's limited scope.

## 3 Experiments

We first introduce the implementation details of our query transformer architecture (§3.1). We then show the empirical performance of our approach compared to state-of-the-art models across 11 benchmarks (§3.2. Finally, we further study the performance-efficiency trade-off (§3.3).

### 3.1 Experimental Setup

**MQT-LLaVA Implementation Details.** We implement our models based on LLaVA-1.5 (Liu et al., 2023a), except that we use our Matryoshka Query Transformer instead of an MLP to obtain the visual tokens. The MQT is a single-layer Transformer with cross-attention. Following Liu et al. (2023a), we select CLIP ViT-L/14 (Radford et al., 2021) as our vision encoder, supporting 336x336 image resolution, and Vicuna-v1.5 (Chiang et al., 2023) as our LLM. As studied in Hu et al. (2023); Zhu et al. (2023); Liu et al. (2023b), we adopt a two-stage training approach. We train only the query transformer in the first-stage alignment, using LLaVA-558K for 1 epoch with a batch size of 256 and a learning rate of 1e-3. We then fine-tune both the query transformer and LLM using LLaVA-665K for 2 epochs with a batch size of 128 and a learning rate of 2e-5. All training is on 8xA6000s, with 4 and 30 hours per stage, respectively. We apply MQT during the second stage (c.f. §4.3).

**Baselines.** As shown in Table 1, we compare our model with LLaVA-1.5 (Liu et al., 2023a) and our model's baseline LLaVA query transformer (QT-LLaVA), which is trained with a fixed number of 256 visual tokens across all training stages. We also list other models' results for comparison, including BLIP-2 (Li et al., 2023a), InstructBLIP (Dai et al., 2023), Shikra (Chen et al., 2023), IDEFICS (IDEFICS, 2023), and Qwen-VL (Bai et al., 2023).

**Evaluation Benchmarks.** We evaluate our model across 11 mainstream benchmarks, including VizWiz (Gurari et al., 2018), ScienceQA-IMG (Lu et al., 2022), VQA-v2 (Goyal et al., 2017), GQA (Hudson and Manning, 2019), POPE (Li et al., 2023c), MME Perception (Fu et al., 2023), MME Cognition (Fu et al., 2023), MMBench (Liu et al., 2023c), LLaVA-Bench (In-the-Wild) (Liu et al., 2023b), and MM-Vet (Yu et al., 2024).

### 3.2 Main Results

Table 1 presents the results of MQT-LLaVA with inference visual token budgets of 2, 4, 8, 16, 36, 64, 144, and 256. We refer to the baseline approach, where the model is trained with a fixed number of visual tokens across all training stages, as LLaVA Query Transformer (QT-LLaVA). MQT-LLaVA

| Method | LLM | Res. | #Tokens | VizWiz | SQA$^I$ | VQA$^{v2}$ | GQA | POPE | MME$^P$ | MME$^C$ | MMMU | MMB | LLaVA$^W$ | MM-Vet | Avg |
|---|---|---|---|---|---|---|---|---|---|---|---|---|---|---|---|
| BLIP-2 | Vicuna-13B | 224 | 32 | 19.6 | 61 | 41.0 | 41 | 85.3 | 1293.8 | – | – | – | 38.1 | 22.4 | – |
| InstructBLIP | Vicuna-7B | 224 | 32 | 34.5 | 60.5 | – | 49.2 | – | 1084 | 229 | 30.6 | – | 60.9 | 26.2 | – |
| InstructBLIP | Vicuna-13B | 224 | 32 | 33.4 | 63.1 | – | 49.5 | 78.9 | 1212.8 | 243 | 33.8 | – | 58.2 | 25.6 | – |
| Shikra | Vicuna-13B | 224 | 256 | – | – | 77.4* | – | – | – | – | – | 58.8 | – | – | – |
| IDEFICS-9B | LLaMA-7B | 224 | 64 | 35.5 | – | 50.9 | 38.4 | – | – | – | – | 48.2 | – | – | – |
| IDEFICS-80B | LLaMA-65B | 224 | 64 | 36.0 | – | 60.0 | 45.2 | – | – | – | – | 54.5 | – | – | – |
| Qwen-VL | Qwen-7B | 448 | 256 | 35.2 | 67.1 | **78.8*** | 59.3* | – | – | – | – | 38.2 | – | – | – |
| Qwen-VL-Chat | Qwen-7B | 448 | 256 | 38.9 | **68.2** | 78.2* | 57.5* | – | 1487.5 | – | – | 60.6 | – | – | – |
| LLaVA-1.5 | Vicuna-1.5-7B | 336 | 576 | 50.0 | 66.8 | 78.5* | **62.0*** | 85.9 | **1510.7** | 316.1 | 34.7 | 64.3 | 63.4 | **30.5** | 59.2 |
| QT-LLaVA | Vicuna-1.5-7B | 336 | 256 | 51.1 | 68.1 | 76.8* | 61.5* | 84.1 | 1431.2 | 348.2 | 34.3 | 64.0 | 63.9 | 27.9 | 58.8 |
| MQT-LLAVA | Vicuna-1.5-7B | 336 | 256 | **53.1** | 67.6 | 76.8* | 61.6* | 84.4 | 1434.5 | **353.6** | **34.8** | 64.3 | **64.6** | 29.8 | **59.4** |
| MQT-LLAVA | Vicuna-1.5-7B | 336 | 144 | 52.0 | 67.5 | 76.4* | 61.4* | 83.9 | 1446.4 | 351.8 | 34.4 | **64.4** | 61.4 | 29.9 | 58.9 |
| MQT-LLAVA | Vicuna-1.5-7B | 336 | 64 | 51.5 | 67.0 | 75.3* | 60.0* | 83.6 | 1464.3 | 352.9 | 34.4 | 63.5 | 59.4 | 28.9 | 58.3 |
| MQT-LLAVA | Vicuna-1.5-7B | 336 | 36 | 51.0 | 66.8 | 73.7* | 58.8* | 81.9 | 1416.3 | 349.3 | 34.4 | 63.4 | 59.6 | 27.8 | 57.4 |
| MQT-LLAVA | Vicuna-1.5-7B | 336 | 16 | 49.8 | 67.5 | 71.1* | 57.6* | 80.8 | 1408.5 | 349.3 | 33.6 | 61.9 | 55.2 | 25.3 | 56.1 |
| MQT-LLAVA | Vicuna-1.5-7B | 336 | 8 | 49.4 | 66.2 | 67.2* | 55.5* | 79.4 | 1282.2 | 323.6 | 33.1 | 58.6 | 51.4 | 21.3 | 53.3 |
| MQT-LLAVA | Vicuna-1.5-7B | 336 | 4 | 49.4 | 65.1 | 64.1* | 53.0* | 77.6 | 1176.1 | 296.8 | 32.8 | 56.5 | 44.3 | 20.2 | 50.8 |
| MQT-LLAVA | Vicuna-1.5-7B | 336 | 2 | 48.5 | 65.0 | 61.0* | 50.8* | 74.5 | 1144.0 | 268.9 | 32.5 | 54.4 | 41.7 | 19.5 | 49.0 |

Table 1: Comparison with state-of-the-art methods on 11 vision-language benchmarks. Our model (MQT-LLAVA) with up to 256 tokens achieves on par or better than LLaVA-1.5 performance across 11 benchmarks, outperforming it on 6 of 11 benchmarks. We mark the best performance in **bold** and the second-best underlined. #Tokens is the number of visual tokens used during inference. Avg is the normalized average across 11 benchmarks, out of 100. Benchmark names are abbreviated for brevity: SQA$^I$: ScienceQA-IMG, MME$^P$: MME Perception, MME$^C$: MME Cognition, MMB: MMBench, LLaVA$^W$: LLaVA-Bench (In-the-Wild). *The training images of the datasets are observed during training.

outperforms the baseline QT-LLaVA with 256 tokens in 9 out of 11 benchmarks. One possible explanation is that by enforcing our model to only see fewer tokens during training, the stricter constraint helps the model generalize better to unseen tasks. This is especially evident in the higher performance on VizWiz. When compared to open-source state-of-the-art models, our model with 256 tokens achieves on par or better than LLaVA-1.5 performance with 576 tokens across 11 benchmarks, outperforming it in 6 out of 11 benchmarks. Even with 64 tokens, our model falls short of LLaVA-1.5 by only 0.9 points on average. When drastically drop to only 2 tokens, our score falls by only 3% on ScienceQA and 6% on MMMU. While directly adding a query transformer to LLaVA degrades the performance, our strategy can achieve comparable or better performance than LLaVA-1.5.

We explore performing inference using a variety of numbers of visual tokens, including 1) an extremely low number of tokens; 2) a number of visual tokens unseen during training. As shown in Figure 3, MQT-LLAVA with only 2 visual tokens outperforms InstructBLIP (Vicuna-7B), which is based on Q-Former (Li et al., 2023a) using 32 visual tokens. This demonstrates the effectiveness of our model in compressing visual information, pointing to its potential use for applications in computation-heavy tasks. For an unseen number of visual tokens, we pick a random number of visual tokens: 77, and include its results in Appendix C. Despite never being explicitly trained for this number of tokens, our model can generalize to any number within 256 during inference, demonstrating a further benefit of our elastic approach.

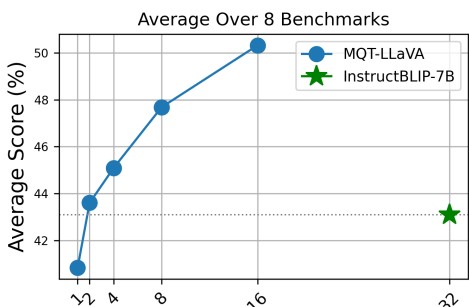

Figure 3: With only 2 visual tokens, MQT-LLAVA outperforms InstructBLIP (which uses 32 visual tokens) on all 8 benchmarks it is evaluated on.

## 3.3 Computational Efficiency

To demonstrate our computational efficiency, we compute TFLOPs when running MQT-LLAVA on MMBench with 8, 16, 36, 64, 144, and 256 visual tokens, compared to LLaVA with 576 tokens. As shown in Figure 1, we are able to achieve significant speed-ups with little-to-no performance loss: our model with 256 and 144 tokens respectively achieve a 2x and 3x speed-up compared to

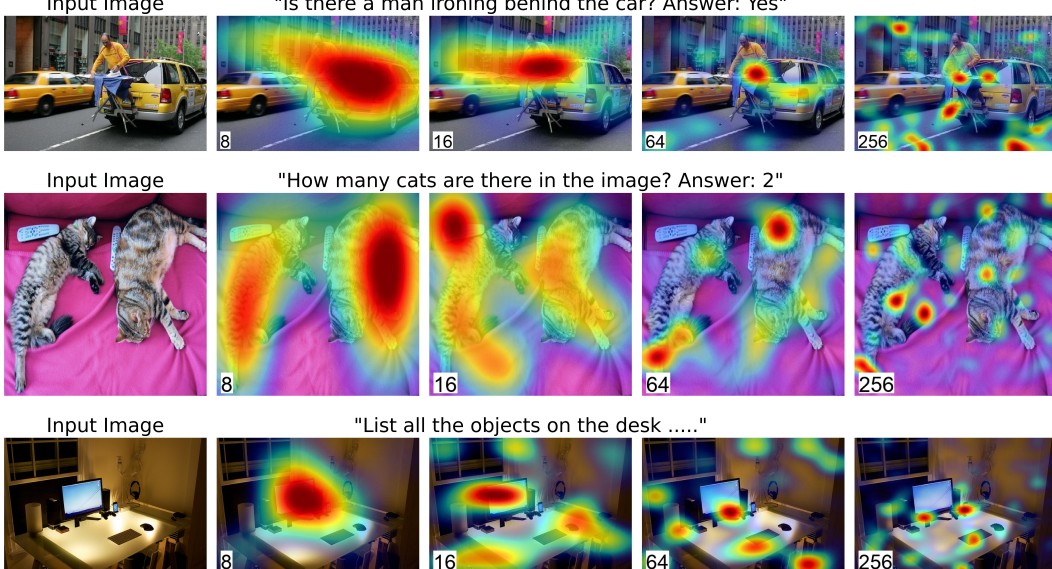

Figure 4: Grad-CAM visualization of 1 randomly picked token from using 8, 16, 64, 256 visual tokens, respectively, to encode an image. The model effectively concentrates on high-level concepts using fewer tokens and delves into low-level details with more tokens. The complete input to the third image is "List all the objects on the desk. The objects on the desk include a computer monitor, a keyboard, a mouse, a cell phone, and a pair of headphones".

LLaVA-1.5 while maintaining the same or even better performance; and when using 16 tokens, we achieve an 8x speed-up with a performance drop of only 2.4 points.

# 4 Analyses

To better understand the meaning of visual tokens and to systematically study the number of tokens required by different vision-language tasks, we investigate two key questions: (1) How does the focus of the model change with varying numbers of visual tokens? (§4.1); and (2) How do different numbers of visual tokens impact various tasks? (§4.2)

## 4.1 How does the focus of the model change with varying numbers of visual tokens?

To explore what visual information each token encodes, we utilize Grad-CAM (Selvaraju et al., 2017) to visualize the focus of visual tokens. As illustrated in Figure 4, we qualitatively analyze the results of using 8, 16, 64, and 256 tokens.

We observe that the model's focus changes with the number of tokens used. When using a few tokens (e.g., 8), the model accurately concentrates on global visual concepts related to the question. As the number of tokens increases (e.g., 256), the model not only attends to the relevant objects but also delves into localized details. For example, in the third image, with 8 tokens, the model focuses on the monitor. With 16 tokens, it includes both the monitor and the mouse. With 64 tokens, it highlights the monitor and keyboard. Finally, with 256 tokens, the model encompasses several objects, including the monitor, keyboard, and cell phone. In the examples from the first and second images, our model effectively focuses on the man ironing behind the car and the two cats, even with only 8 tokens. The impressive qualitative results, especially those using only a few tokens, demonstrate the effectiveness of our approach and the strong capabilities obtained despite using a minimal number of tokens.

## 4.2 How do different numbers of visual tokens impact different tasks?

When using varying numbers of visual tokens during inference, we observe that the model's performance change varies across different tasks.

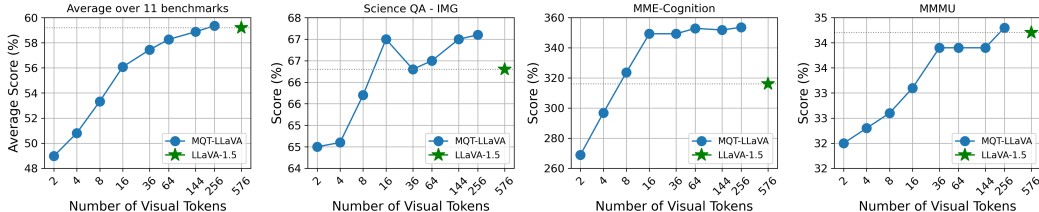

Figure 5: The number of visual tokens impact different tasks differently (x-axis is in log-scale). Our model's performance on ScienceQA, MME-Cognition and MMMU is remarkably robust to token reduction. For full visualization of all 11 benchmarks, see Figure 8 and Figure 9 in Appendix.

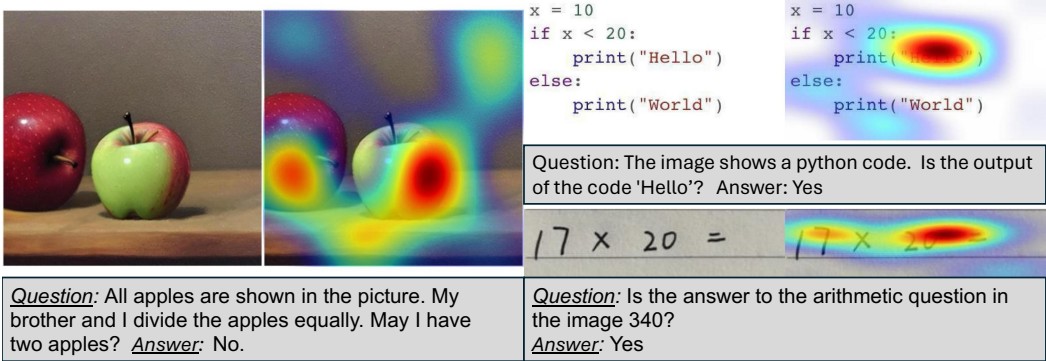

Figure 6: Examples from MME Cognition. Grad-CAM results are from using 16 tokens which answered all the questions correctly.

**Tasks requiring a large number of visual tokens.** Tasks that require fine-grained visual understanding and deep reasoning across multiple areas of the image naturally demand a higher number of visual tokens for optimal performance. When the number of visual tokens decreases, the encoded image information is reduced, leading to performance degradation. This trend is evident in tasks such as VQAv2, GQA, VizWiz, MMBench, LLaVA-Bench, and MM-Vet. As illustrated in Appendix Figure 8, the performance on these tasks gradually declines as the number of visual tokens decreases from 256, with a more rapid decline observed when the tokens are further reduced.

**Tasks robust to visual token reduction.** In contrast, for several benchmarks primarily targeting the visual perception skills of models, performance remains consistent when gradually reducing the number of visual tokens until a threshold is reached. Beyond this threshold, performance drops significantly (see Figure 5 and Appendix Figure 8). This "turning point" is observed in benchmarks such as MME Cognition, MME Perception, POPE, and MMMU.

For instance, in MME-Cognition (see Figure 6), tasks involving commonsense reasoning, code reasoning, and numerical calculation can be performed effectively with as few as 16 visual tokens, allowing the model to focus on the relevant image sections. Similar results are seen in other tasks, like the hallucination question "Is there a car in the image?" from POPE. However, once the "turning point" is reached, further reducing the number of visual tokens prevents the model from attending to the correct objects, leading to a sharp decline in performance.

Another notable observation comes from ScienceQA and MMMU, which contain subject-specific questions from school curricula. The model's performance on these tasks remains robust despite a decrease in visual tokens, achieving scores of 65.0 and 32.5, respectively, with only 2 tokens. This suggests that the reasoning required for academic questions is primarily conducted by the language model (LLM); even with minimal visual hints, the LLM can interpret the image content and perform the reasoning tasks effectively.

**When are fewer visual tokens better?** As shown above, MQT-LLAVA with 16 tokens can achieve better performance on ScienceQA compared to MQT-LLAVA with 144 tokens. To understand

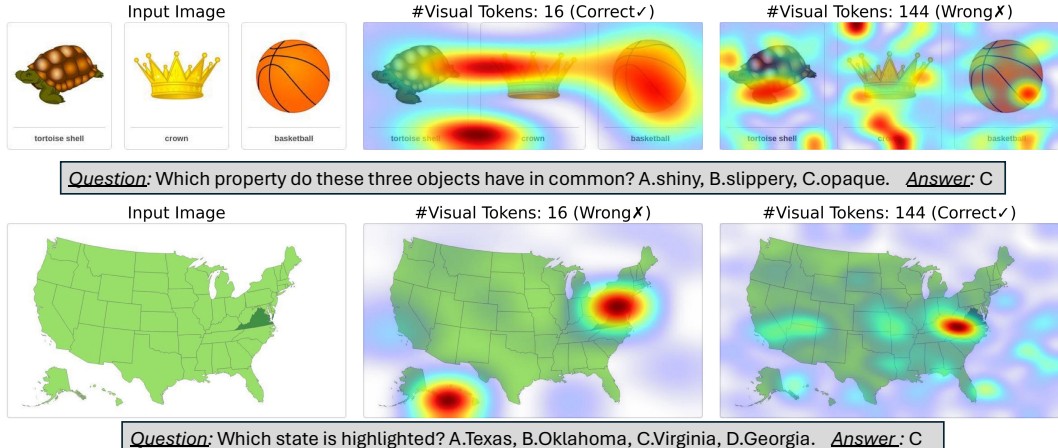

Figure 7: Comparison of correct and failure cases in 16 vs 144 visual tokens on Science-QA (test-set).

| Method | VisWiz | SQA$^I$ | VQA$^{v2}$ | GQA | POPE | MME$^P$ | MME$^C$ | MMMU | MMB | LLaVA$^W$ | MM-Vet | Avg |
|---|---|---|---|---|---|---|---|---|---|---|---|---|
| QT-LLaVA (Baseline) | 51.1 | **68.1** | **76.8**$^*$ | 61.5$^*$ | 84.1 | 1431.2 | 348.2 | 34.3 | 64.0 | 63.9 | 27.9 | 58.8 |
| MQT-LLAVA (Ours) | **53.1** | 67.6 | **76.8**$^*$ | **61.6**$^*$ | **84.4** | **1434.5** | **353.6** | **34.8** | **64.3** | **64.6** | **29.8** | **59.4** |
| w/ Log-based Matryoshka Tokens | 51.2 | 67.4 | 75.6$^*$ | 60.3$^*$ | 83.2 | 1418.9 | 314.1 | 32.8 | 62.6 | 59.2 | 27.3 | 57.3 |
| w/ Project then Attention | 50.5 | 66.8 | 73.4 | 57.1 | 82.3 | 1382.8 | 317.5 | 32.7 | 61.4 | 60.0 | 29.5 | 56.6 |
| w/ First-stage training with Query Transformer | 51.6 | 67.2 | 75.9$^*$ | 60.5$^*$ | 82.6 | 1378.6 | 295.4 | 33.2 | 63.1 | 56.5 | 26.8 | 56.7 |

Table 2: For simplicity in ablation studies, we evaluate all the models with 256 visual tokens. All models are trained with the same hyperparameters.

why fewer tokens may benefit this task, we qualitatively analyze instances where MQT-LLAVA succeeded with 16 visual tokens, but failed with 144. We show a representative example in Figure 7. MQT-LLAVA with 16 visual tokens attends to all three objects, allowing it to understand their mutual relationship and answer the question correctly. On the other hand, with 144 visual tokens, MQT-LLAVA focuses on various portions of the image and attend to each object independently. This discourages the model from reasoning with the common attributes among the three objects, thus predicting the wrong answer. In summary, fewer visual tokens seems to be preferable when fine-grained visual understanding is not required.

However, it should be noted that using fewer tokens is not always better in this case. As shown in Figure 7, MQT-LLAVA with 144 tokens precisely identified state of Virginia on the map and answered the question correctly. Whereas 16 tokens concentrated on another region which potentially confused its final prediction, lacking the abilities of distinguishing local details of the geographic shape on the map.

## 4.3 Ablation Studies

We ablate several design choices across 11 benchmarks in Table 2. Each ablation independently modifies our best variant, MQT-LLAVA, to create new variants. *(i) linear vs. log-based token number selection*. We replace our linear growth elastic tokens, i.e., $m \in \{2, 4, 6, \ldots, 252, 254, 256\}$ to the log-based approach of MRL, i.e., $m \in \{2, 4, 8, 16, \ldots, 128, 256\}$. This results in an average accuracy of 57.3%, 2.1% lower than MQT-LLAVA, validating our hypothesis that gradually compressing the visual tokens helps the model perform better than log-based choices. *(ii) query transformer architecture*. As mentioned in §2, we choose to first perform cross-attention between query tokens and visual features, then project the learned visual tokens to the LLM. We call this technique "attention then projection". The alternative variant is "projection then attention", which achieves lowest average performance, with a score of 56.6%. This suggests that directly applying the attention mechanism helps preserve the rich grid features, making them better projected to the LLM. *(iii) first-stage pretraining with query transformer*. As mentioned in §3.1, we choose to apply our elastic training paradigm only during the second stage. Experimental results demonstrate that adopting elastic training during the first stage leads average performance dropped by 2.7%. We hypothesize that the first stage aims to align the randomly initialized query tokens with vision-language awareness.

Therefore, it is important to train all 256 tokens with this prior knowledge before reducing the number of tokens in the second stage.

## 5 Related Work

### 5.1 Large Vision Language Model

Large Vision Language Models (LVLMs), pioneered by Liu et al. (2023b); Zhu et al. (2023); Dai et al. (2023); Yin et al. (2023) have successfully showcased promising results on a wide variety of vision-language perception and reasoning tasks. Recent works further expand the capabilities of LVLMs to region-level image understanding (Huang et al., 2023; Peng et al., 2023; Chen et al., 2023; Cai et al., 2024a), video understanding (Zhang et al., 2023; Li et al., 2023b; Jin et al., 2024) and 3D understanding (Hong et al., 2023; Szot et al., 2024). These models mostly consist of a vision encoder and an LLM aligned by a vision-language connector module, which can be an MLP (Liu et al., 2023b; Hu et al., 2023), Q-Former or queries through cross attentions (Dai et al., 2023; Bai et al., 2023; Ye et al., 2023), or Resampler (Alayrac et al., 2022). The number of visual tokens output by these modules can be very large, especially for higher image resolutions, multiple frames in video tasks, and multiple images for in-context learning. In this paper, we propose a training paradigm that can dynamically choose a number of visual tokens that adapts to variable computation costs at inference time.

### 5.2 Efficient Vision Transformers

Reducing the computational cost of LVLMs at deployment is an active area of research. Several works, e.g., TinyLLaVA (Zhou et al., 2024) and LLaVA-Phi (Zhu et al., 2024), reduce the size of the LLM backbone by replacing it with a smaller one, e.g., Phi-2 (Javaheripi et al., 2023). MobileVLM (Chu et al., 2023) and MobileVLM-v2 (Chu et al., 2024) focus on a compact architecture design and training paradigms specifically for mobile usage. In these cases, the computation reductions come from reducing the size of either the LLM or vision encoder backbones, whereas our method focuses on increasing LVLM efficiency by dynamically reducing the number of visual tokens.

A long-standing issue with vision transformers is that the attention mechanism introduces computational complexity that scales quadratically with the input tokens. Vision Longformer (Zhang et al., 2021) adopts sparse attention (Kitaev et al., 2020) to speed up vision transformers for larger inputs. Other works design various strategies to retain the most informative tokens and reduce the number of visual tokens at the inference stage (Fayyaz et al., 2021; Liang et al., 2022; Rao et al., 2021; Bolya et al., 2023; Yin et al., 2022). Most similar to our work is SparseFormer (Gao et al., 2024) which employs cross-attention to learn sparse representations of both latent tokens and RoI descriptor tokens. In this work, we use the most simple query transformer architecture to actively control the number of visual tokens and explore the impact of reducing the number of visual tokens in LVLMs.

Concurrent work (Cai et al., 2024b) also studies matryoshka-style visual tokens, and designs 5 scales of pooling layers to control the granularity of images. Different from their work, we introduce a query transformer to extract visual tokens, enabling a more flexible choice of any number of visual tokens under a predefined maximum. Their work corroborates our findings by demonstrating robust performance and efficient use of a minimal number of visual tokens.

## 6 Conclusion

In this work, we present MQT-LLAVA, a single vision-language model that enables elastic inference on various downstream tasks and computation resources. We demonstrate that our model achieves performance comparable to or better than training with a fixed number tokens. MQT-LLAVA matches the performance of LLaVA-1.5 across 11 benchmarks using less than half the number of visual tokens, and outperforms LLaVA-1.5 in 6 out of 11 benchmarks. We achieve an 8x less TFLOPs when reducing to 16 tokens while only sacrificing the performance on MMBench by 2.4 points. We hope our exploration of the trade-off between the accuracy and computational cost caused by the number of visual tokens will facilitate future research to achieve the best of both worlds.

## Acknowledgments

We would like to thank members of UCLA NLP and PLUS Lab for their helpful comments. We also thank the reviewers for their valuable reviews. This work was supported by an Amazon gift grant, ONR grant N00014-23-1-2780, U.S. DARPA ECOLE Program No. #HR00112390060, and an SRA from Optum Labs. The views and conclusions contained herein are those of the authors and should not be interpreted as necessarily representing the official policies, either expressed or implied, of DARPA, or the U.S. Government. The U.S. Government is authorized to reproduce and distribute reprints for governmental purposes notwithstanding any copyright annotation therein.

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

# A  Broader Impact

The deployment and release of MQT-LLAVA carry both potential benefits and risks. These considerations include visual aspects as well as common issues found in existing LLMs like Alpaca and Vicuna. Since MQT-LLAVA is built on LLaMA, Vicuna, and CLIP, it inherits certain challenges associated with LLMs and vision encoders.

**Hallucination**   Similar to other LLMs, MQT-LLAVA might produce outputs that are not based on factual information or input data. This raises concerns about the accuracy of inferences, particularly in critical applications such as medical fields.

**Biases**   Biases present in the base models can be brought to MQT-LLAVA, stemming from both the vision encoder (CLIP) and the language decoder (LLaMA/Vicuna). This may result in biased outcomes or unfair representations of diverse content.

**Energy Consumption**   We followed LLaVA-1.5's datasets (smaller datasets compared to other methods) in training our model, which makes energy consumption is not a primary concern.

# B  Limitation

Despite our remarkable performance, one limitation of our work is that the maximum number of tokens MQT-LLAVA can accommodate at inference time is 256. We leave the exploration of using larger numbers in inference than training time to future work.

# C  Additional Results

We include how number of visual tokens impact the different tasks differently across all 11 benchmarks in Figure 8 and Figure 9

We present the results of choosing a random number of visual tokens, 77 as shown in Table 3, to demonstrate our flexibility in selecting any number of tokens during inference.

To demonstrate that the visual tokens used for visualization in Figure 4 are not cherry-picked, we present all the first eight tokens in Figure 10.

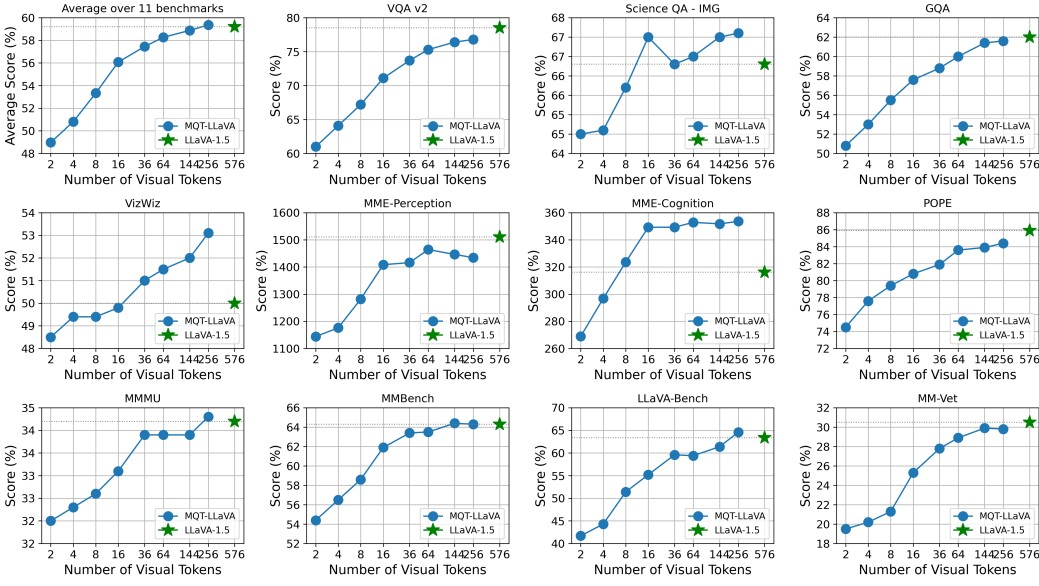

Figure 8: The number of visual tokens impact the different tasks differently across 11 benchmarks. We log scaled x-axis for readability.

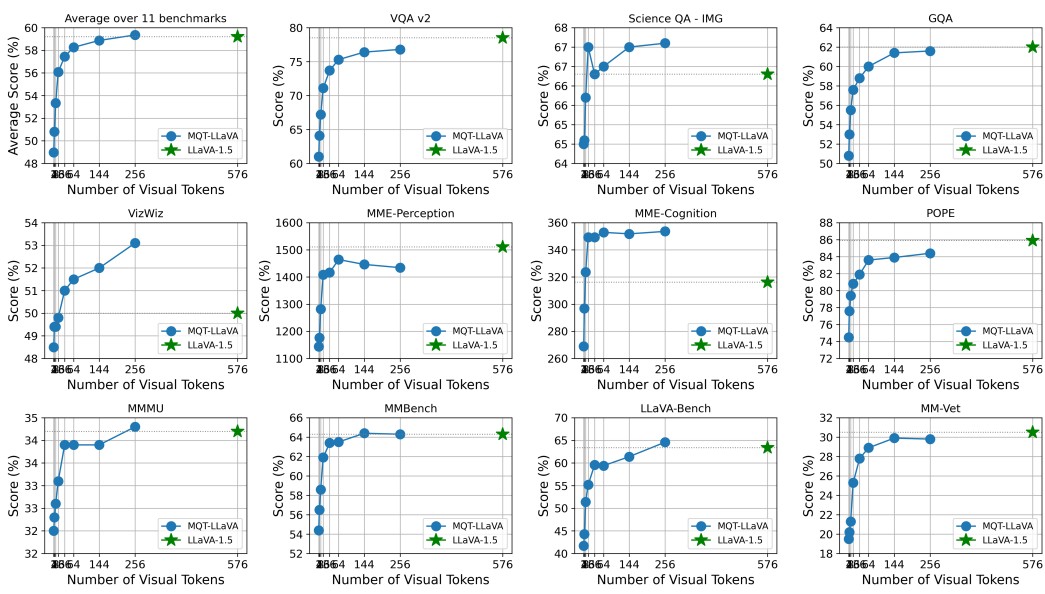

Figure 9: The number of visual tokens impact the different tasks differently across 11 benchmarks. No log scaled on the x-axis is applied.

| Method | LLM | Res. | #Tokens | VizWiz | SQA$^I$ | VQA$^{v2}$ | GQA | POPE | MME$^P$ | MME$^C$ | MMMU | MMB | LLaVA$^W$ | MM-Vet | Avg |
|--------|-----|------|---------|--------|---------|------------|-----|------|---------|---------|------|-----|-----------|--------|-----|
| QT-LLaVA | Vicuna-1.5-7B | 336 | 256 | 51.1 | **68.1** | **76.8**$^*$ | 61.5$^*$ | 84.1 | 1431.2 | 348.2 | 34.3 | 64.0 | 63.9 | 27.9 | 58.8 |
| MQT-LLAVA | Vicuna-1.5-7B | 336 | 256 | **53.1** | 67.6 | **76.8**$^*$ | **61.6**$^*$ | **84.4** | 1434.5 | **353.6** | **34.8** | 64.3 | **64.6** | 29.8 | **59.4** |
| MQT-LLAVA | Vicuna-1.5-7B | 336 | 144 | 52.0 | 67.5 | 76.4$^*$ | 61.4$^*$ | 83.9 | 1446.4 | 351.8 | 34.4 | **64.4** | 61.4 | **29.9** | 58.9 |
| MQT-LLAVA | Vicuna-1.5-7B | 336 | 77 | 51.6 | 67.1 | 75.8$^*$ | 60.4$^*$ | 83.6 | 1457.0 | 336.1 | 34.0 | 64.0 | 59.9 | 29.3 | 58.3 |
| MQT-LLAVA | Vicuna-1.5-7B | 336 | 64 | 51.5 | 67.0 | 75.3$^*$ | 60.0$^*$ | 83.6 | **1464.3** | 352.9 | 34.4 | 63.5 | 59.4 | 28.9 | 58.3 |

Table 3: Results of MQT-LLAVA with different numbers of visual tokens. To demonstrate our flexibility in selecting any number of tokens up to 256, we chose a random number of visual tokens during inference, 77, which was not seen during training.

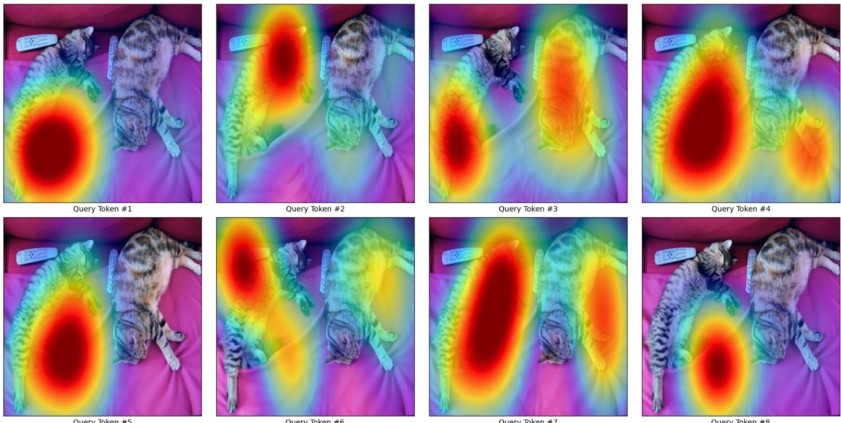

Figure 10: Grad-CAM visualization from all the tokens in our model when inference with 8 tokens. Input: "How many cats are there in the image? Answer: 2".

