# OpenReview forum: "Matryoshka Query Transformer for Large Vision-Language Models"
_NeurIPS.cc/2024/Conference — NeurIPS 2024 poster_

### Official Review · Reviewer_L7Jp · 2024-06-15

**Soundness:** 4
**Presentation:** 4
**Contribution:** 3
**Rating:** 7
**Confidence:** 5

**Summary:**

This paper introduces a new concept called Matryoshka Query Transformer (MQT) that brings in the concept of Matryoshka information packing to make visual tokens flexible and can be used in multimodal vision language models like LLaVA. The reduced tokens tend to reduce the quadratic complexity of the language model due to potential flexibility in the processed visual tokens.

Owing to this, there are significant efficiency benefits all the while retaining accuracy across various benchmarks. The paper also showcases analysis and further discussion on the technique.



----------------------------
The review will be short and does not reflect the time put in for the review or the quality of the paper. When the ideas are simple and clear -- I tend to write shorter reviews to the point.

**Strengths:**

I will go sequentially

1) The paper is extremely well-written and easy to follow.
2) The core ideas, while not completely novel as mentioned by the authors, Matryoshka and the Query transformer, making them work together and showing incredible benefits is commendable and is a worthy contribution.
3) The modelling very clear and the mechanism to achieve and the details are very well fleshed out.
4) The experiments are extensive, with a good analysis.
5) Great job with visualizations and nice to see some TFLOPs measurement.
6) Good analysis beyond benchmark numbers.

Overall, this is a solid work with practical utility. I have a few questions about the paper mentioned in the weaknesses and would appreciate answers for them in the rebuttal, however, I am happy with the paper and am willing to champion it unless there is something I am missing found by other reviewers.

**Weaknesses:**

Most of these are not weaknesses, but rather questions

1) How are you measuring TFLOPs? What does it include, the cost of vision encoder? LLM processing and generation? I might have missed this in the paper and any pointer would be great.
2) While I understand the when you are picking a fixed # token for MQT, you make that choice before in hand. What happens if you actually just obtain 256 tokens and take the first 8? What will be the performance in that case? I know softmax etc make huge differences, but would be good to see that.
3) Any thoughts on sampling vs joint training?
4) Why is log-based (actually exponential) spacing so poor (not 3.5% as mentioned in the paper but 2.1% from the table) so much worse than fine granularity? This is a bit surprising to me because 256 token performance should not be affected because of the lower token counts.
5) In visualizations, what do you mean by one random token?

I am looking forward to your answers on these things.

**Questions:**

See above.

**Limitations:**

Yes

---

> ### Author Rebuttal · Authors · 2024-08-07
>
> Thank you for your review!
>
> 1. Measure of TFLOPs
>
> We measured the TFLOPs by incorporating everything, i.e, including the cost of vision encoder and LLMs. Thank you for pointing out, we would like to incorporate this detail in our revised version.
>
> 2. Scenario of just obtain 256 tokens and take the first 8.
>
> During the inference process, the inputs to MQT are queries and visual features from vit, then the cross attention mechanism will output the learned visual feature before feeding into LLM. So there is no softmax involved in this process. We also experimented to find out that indeed no performance changed in this scenario.
>
> 3. Sampling vs joint training.
>
> For joint training, we can choose multiple $m$ tokens jointly. We think this design can slightly increase the computation of each training step, but may optimize faster from a global steps view. It’s also interesting to consider the design of how and where to choose the multiple $m$ tokens, for example choose $m$ from a local region (i.e, each $m$ is close with each other) or from a more global region (i.e, joint the the first token and all the 256 tokens). Thanks for your suggestion, we will leave this exploration to our future work.
>
> 4. Why log based is worse than fine-grained, not 3.5% as  mentioned in the paper but 2.1% from the table.
>
> We apologize for this confusion. We think the 2.1% absolute performance drop which is observed from the table is the same as 3.5% mentioned in the paper, which represents the relative performance change compared to our method. 3.5% is (our score (59.4) - abalated model score (57.3)) / 59.4. We only employed this relative score comparison in our ablation study table. We will incorporate this explanation into our revised version. As we discussed in line 249, this validated our hypothesis that gradually compressing the visual tokens helps the model perform better than log-based choices. Although the tested scenario is 256 tokens, the first few tokens of 256 are still “log-based” trained in the ablated model (for example, still large gaps between 64 and 128 tokens or 128 and 256 tokens), which experimentally found to be not as good as our final model.
>
> 5. How is a random token picked in the visualization?
>
> For example, we provided the visualization when our model inference with 8 tokens, please refer to Figure 1 in our general response. We will also incorporate this visualization to our revised version. Here, say we pick the 6th token which is on the 0.75 percentile location. For other settings such as 16 tokens or 256 tokens, we fixed it to randomly pick one visual token for visualization purposes.

---

> > ### Comment · Reviewer_L7Jp · 2024-08-07
> >
> > Thanks for the rebuttal.
> >
> > 1) Good to know about what was being measured in TFLOPs
> >
> > 2) Can you please share the results of the experiment of picking first 8 tokens?
> >
> > 3) Sounds good about joint vs sampled
> >
> > 4) I would prefer having aboslute numbers than relative to reduce confusion. But I am still suprised by the performance gap here. But I can see why it might happen.
> >
> > 5) Thanks for the information. Please add this to the main paper as promised.
> >
> > I shall look at other reviews, rebuttals and discssion before making a final decision, but I lean to accept this paper even in the current stage.

---

> > > ### Author Response · Authors · 2024-08-07
> > > **Thank you for the discussion**
> > >
> > > Dear reviewer, Thank you for your promptly and insightful feedback. We appreciate your lean to accept this paper.
> > >
> > > 1. Thanks.
> > > 2. The results of inference with the first 8 tokens are in our paper's Table 1. We have tested your proposed scenarios and found to be exact the same numbers.
> > > 3. Thanks.
> > > 4. Thank you for your advice, we realized this confusion, too. We will revise this expression in line 249 and 258 to absolute scores.
> > > 5. Thanks, we will definitely add this visualization to the main paper.

---

### Official Review · Reviewer_k14Y · 2024-07-13

**Soundness:** 2
**Presentation:** 2
**Contribution:** 2
**Rating:** 6
**Confidence:** 4

**Summary:**

Authors try to use Matryoshka mechanism to guide the learning process of LVLMs such as llava.

Authors show a pretty good scaling curve with different amount of visual tokens.

**Strengths:**

1. The idea is interesting.
2. The presentation is clear.
3. The attention visualization is interesting.

**Weaknesses:**

1. For a given image, which scale should I choose to achieve best trade off? Is there an answer?
2. Authors use more compute (2 epoch) to get the similar performance as LLaVA, is there any reason for this?
3. All designs are center upon the qformer. There is little study on this. How many layers do we need? Do we really need it?

**Questions:**

1. What are the performance under other datasets like TextVQA and ChartQA? Those are dataset different from the natrual image distribution.
2. The visualizations are good but we don't have quantitive numbers to support your claim. Since those pictures can be cherry picked.

**Limitations:**

See comments above.

---

> ### Author Rebuttal · Authors · 2024-08-07
>
> Thank you for your review!
>
> 1. What scale to choose for the best trade-off?
>
> The trade-off visualization is from Figure 5 and the complete version is from Figure 8 in our paper’s appendix. For the best tradeoff, as we mentioned in line 215, we observed a ''turning point'' in many benchmarks. This point has a relatively lower computation but keeps a high performance. After this point, the performance will decrease more drastically although with a much lower computation. We think this point can be a good answer for the best trade-off.
>
> 2. Training Computation
>
> Our main contribution is to achieve a single model for elastic inference of **any** visual tokens (line 7-9 in our Abstract). As mentioned in our methods (see line 102), we achieved this by choosing $m$ from {2, 4, 6, …, 252, 254, 256}, which means only the first 2 tokens are always supervised for the entire training data, whereas for the rest of visual tokens, less supervision is applied, especially for the visual tokens at the tailing positions. But our gradually reducing token numbers strategy helps stabilize this issue as we demonstrated robust performance in our experiments. Therefore, we apply 2 epochs of training data to help this issue. Gladly, LLaVA instruction tuning is very computational friendly. We only trained for 15 hours (one epoch) more on one A6000 machine, which is a cost with less concern.
>
> 3. Study on Q-Former
>
> Concerning the Q-Former layers. We studied the layers for MQT transformers at the early stage of our research. We didn’t find improved performance with multiple layers. This similar design is also employed by Qwen-VL (Bai et al., 2023) which is a strong LVLM in the open-source community. Our extensive experiments in Table 1 and comparison to Q-Former in Figure 3 also demonstrated the sufficient capabilities of MQT design. We also incorporated LLaVA + vallina Q-Former results in the Table 1 of our general response.
>
> Concerning Q-Former at all. Our MQT design is to achieve a single model which is trained only once for elastic inference of **any** visual tokens (line 7-9 in our Abstract). Query Transformer design has a learned number of visual tokens, therefore, during inference, an elastic number of visual tokens can be choosed for various downstream tasks to achieve our goal.
>
> 4. TextQA and ChartQA
>
> Please refer to the general response’s Table 2 for added TextVQA results. We observe a very steady performance trend even when reducing the number of visual tokens to 16 tokens. In our paper’s Table 1, we have tested on 4 comprehensive evaluation benchmarks for Large Multimodal Models including MME,  MMMU, MMBench, and MM-Vet, where our model demonstrated robust performance on these tasks. All 4 benchmarks involving evaluation on OCR and text related tasks, we report the specific scores for these sub categories in our general response’s Table 3, please refer to it.  In summary, our model performs the best in these text related tasks with 3.2 points higher than LLaVA-1.5 on average.
>
> As for ChartQA, since LLaVA-1.5 didn’t incorporate any training data for chart understanding, their paper didn’t evaluate on this dataset. When we downloaded their weight and conducted evaluation, it reveals LLaVA-1.5 performs very poorly on ChartQA. Therefore, this task may not be suitable for use as a reference for comparison. But we also provided these results here for your reference. LLaVA-1.5 with 576 tokens achieved 18.2 on ChartQA, and our model with only 256 tokens, reducing to  [144, 64, 36, 16, 8, 4] achieved 14.3,  14.1,  14.0 , 13.6 , 13.7 , 12.7 , and 12.0 separately. This also indicates the strong robustness of our model in reducing numbers of visual tokens.
>
> 5. Attention map visualization and quantitative results.
>
> We have conducted extensive quantitative evaluation in our paper’s Table 1 and Table 2 and demonstrated strong capabilities of our model. We conducted attention map visualization in order to understand what visual information is the model focusing on when using a lower number of visual tokens. The visualization on 1 visual token is **not cherry-picked** but randomly picked. For example, we provided the visualization when our model inference with 8 tokens, please refer to Figure 1 in our general response. We will also incorporate this visualization to our revised version. For other settings such as 16 tokens or 256 tokens, we fixed it to randomly pick one visual token for visualization purpose.

---

> > ### Comment · Reviewer_k14Y · 2024-08-10
> > **Reply to authors**
> >
> > Thank you authors for providing the rebuttal.
> > I am still doubtful about the visualizations and even overall the story line provided. The supervision signal under different granularities of visual tokens are from the same set of captions. How can we make sure this makes sense and model really learns the correct granularity? If so, then the assumption about the visualizations do not make sense.

---

> > > ### Author Response · Authors · 2024-08-11
> > > **Reply to the question on correct granularity.**
> > >
> > > Dear reviewer, Thank you for your feedback. As demonstrated in the original Matryoshka Representation Learning (MRL) work (Kusupati et al., NeurIPS 2022), the model can learn the correct granularity even when the supervision signals come from the same source. As mentioned in MRL work, this coarse-to-fine granularity is achieved by explicit optimization of $O(log(d))$ lower-dimensional vectors in a nested fashion.
> > >
> > > Similarly in our work, although the supervision signal remains consistent, our model is trained with nested varying numbers of visual tokens, allowing it to adapt to different settings. We acknowledge that the model may learn the correct granularity implicitly, even though we employed an explicit variation in the number of visual tokens. Thus, to support this, we provide both quantitative data (Table 1) and qualitative visualizations (Figures 4, 5, and 7) demonstrating that this implicit training is effective and that the learned patterns are intuitive.
> > >
> > > We hope our response can address your question. Thank you.

---

### Official Review · Reviewer_jmsV · 2024-07-13

**Soundness:** 3
**Presentation:** 3
**Contribution:** 2
**Rating:** 6
**Confidence:** 3

**Summary:**

The paper considers multimodal vision transformers, in which we have a stream of both visual and textual tokens. Current architecture typically assume a fixed number $m$ of visual tokens. In contrast, the proposed `MQT` aims to achieves a dynamic number of visual tokens. This is done by integrating a form of compression+dropout mechanism during training. Specifically, the image tokens are compressed to $M$ tokens using cross-attention. To vary the number of tokens, the model only needs to use only  $m < M$ queries from te cross-attention mechanism. At training time, this is done by randomly selecting $m$ and feeding the **first** $m$ queries.

**Strengths:**

* The proposed method is a simple idea that can be added as plug-and-play to any model
* generally the idea of using some form of dropout also often has benefits for regularization so it may also benefits more than efficiency
* The paper is well written and has comprehensive ablation experiments

**Weaknesses:**

- **Lack of comparison with dynamic baselines**: in the field of image-only ViT, there is a flourishing literature on dynamically reducing the number of tokens using merging or pruning or scale selection. I am not as familiar with this literature for multimodal models, but there does seem to be similar existing approaches: for instance, *LLaVA-PruMerge: Adaptive Token Reduction for Efficient Large Multimodal Models* is one, though it might be too recent for the authors to have considered it for the submission. However, more generally, having a simple off-the-shelf token merging/pruning approach for the ViT encoder would be interesting.

- **No improvement over baselines in the low FLOPs regime**. From Figure 1, it seems that the benefits of the proposed `MQT` decrease in the low FLOPs regime, when increasing the sparsity ratio of $M / m$.  This limits the usefulness of the methods in the realm of model efficiency, though it seems to be a useful and simple training strategy for the high FLOPs regime.

**Questions:**

- How are the $m$ latent queries chosen at inference ? I am assuming $m$ is chosen as a hyperparameter, and the first $m$ queries are kept, however it is not very clear from the methods section

- Addressing the **low FLOPs regime**:   From Figure 1, it seems that the benefits of the proposed `MQT` decrease in the low FLOPs regime, when increasing the sparsity ratio of $M / m$. I think it is not necessarily a surprising result as increasing sparsity too much will severely impact accuracy. However, I am wondering how would the trade-off curves would look like when starting with a base `QRT` model with a smaller $M$ and lower sparsity ratio, i.e. with a model that would achieves same number of tokens/FLOPs, but where the relative sparsity ratio to achieve is less aggressive.

**Limitations:**

The paper has a limitation section but in my opinion it does not fully address some limitations of the paper (e.g. no improvement in performance in the low FLOPs regime)

---

> ### Author Rebuttal · Authors · 2024-08-07
>
> Thank you for your review!
>
> 1. Comparison with dynamic baselines.
>
> For comparison with LLaVA-PruMerge, please refer to our Table 1 in general response. In summary,
> Our model significantly outperformed LLaVA-PruMerge in **5 out of 6 tasks**, and by a **2.3** absolute score on average. Moreover, LLaVA-PrugeMerge employs a interquartile range (IQR) method to identify import visual tokens, the identified number is “fixed” in IQR. Whereas our MQT can enable a dynamic number of **any** visual tokens during inference.
>
> As for dynamic in ViT encoder, the following discussion is directly cited from LLaVA-PruMerge paper Section 3.5 which corresponds with our thoughts as well. ''Block by block, tokens are gradually reduced in number, correspondingly reducing the computation cost in the internal ViT'',  ''LMMs usually require a large stack of visual tokens… Thus, using previous token merging methods to obtain one refined class token as representation of visual input is not consistent with the literature of large multimodal models''.
>
> To the best of our knowledge, we didn’t find a dynamic visual tokens baseline in ViT for us to compare. Besides, training a dynamic ViT for this purpose is very computationally intensive. This highlights the efficiency and lightweight nature of our MQT design.
>
> 2. Improvement in Low FLOPs regime.
>
> Our main contribution is not to train a model which performs better than independently trained low FLOPs models. But rather to have a model that can train only once that can enable elastic inference of **any** number of visual tokens. Please refer to line 12-15 in our abstract and line 43-44 in our introduction section. The illustration of Figure 1 demonstrated that our model is at least as good as training (exhaustively 256) possible models by picking several choices of visual token numbers in this range for evaluation and comparison. And we are glad to find out that our model even achieved better performance in high FLOPs regime.
>
> 3. How is $m$ chosen at inference? from the method section?
>
> Yes, your assumption is correct. Please also refer to line 50 in our introduction section: ''During inference, we have the flexibility to selectively utilize solely the initial $m$ visual tokens''. In the method section, we mentioned how $m$ is chosen for training time such as at line 99, therefore how $m$ is chosen at inference time is intuitive. We will incorporate this detail to the method section in our revised paper. Thank you.
>
> 4. Addressing Low FLOPs regime and MQT model starting with a lower $M$
>
> Please first see the response to (2) above. For your question about the base MQT starting with a smaller $M$ and lower sparsity ratio, we believe you are probably referring to the baseline model with ''Fixed #Visual Tokens'' as illustrated in our Figure 1. Its results are indicated by the orange curve. If we misunderstand or you have further questions, please respond to our rebuttal and we are very happy to address your questions.

---

> > ### Comment · Reviewer_jmsV · 2024-08-11
> >
> > Dear authors,
> >
> > Thanks for your response. I do agree that elastic inference is a beneficial property, which MQT seems to reach with a simple training strategy. Although I still think the efficiency/accuracy trade-off of the method could be investigated a bit more in-depth, in particular for different number of baseline tokens.  (see below).
> > However, since I do not have any strong negative left after the rebuttal, I will raise my score to weak accept.
> >
> > **Regarding 2/4:** My suggestion was rather on trying different sparsity ratio vs baseline number of tokens trade-off, to see how this affects the performance in the lower FLOPs regime:
> > Taking Figure 1 as an example, it is my understanding that the blue curve is a model trained with 576 tokens (baseline) + MQT to enable elastic inference with a lower number of tokens. so for instance, to achieve 64 tokens at inference a sparsity ratio of ~10% is applied. But another way to attain 64 tokens at inference would to use a 256 token baseline model (*which seems to only be roughly $0.2$ point of accuracy below the 576 tokens one in Figure 1*) with a less drastic sparsity ratio of 25%, which may hurt accuracy less and lead to a better trade-off curve.

---

> > > ### Author Response · Authors · 2024-08-11
> > > **Thank you for the discussion**
> > >
> > > Dear reviewer, Thank you for your feedback. We are encouraged by your decision to raise the score to accept this paper. We agree that the efficiency and accuracy trade-off method can be investigated more in-depth. As you mentioned, we demonstrated elastic inference and compared it with independently trained baselines to support our "beneficial property". We will leave a more in-depth exploration of the efficiency and accuracy trade-off to future work. Thank you!

---

### Official Review · Reviewer_gY5g · 2024-07-15

**Soundness:** 3
**Presentation:** 3
**Contribution:** 3
**Rating:** 5
**Confidence:** 5

**Summary:**

This paper addresses the challenge of achieving flexibility in the number of visual tokens to suit different tasks and computational resources. Inspired by Matryoshka Representation Learning (MRL), the authors propose the Matryoshka Query Transformer (MQT), which allows for any number of visual tokens during inference. Experimental results demonstrate considerable performance across varied visual token lengths and show promising results even with extreme token numbers.

**Strengths:**

1. The problem of handling an arbitrary number of visual tokens is highly valuable. The authors propose a novel MRL-based method to tackle this issue.

2. The experimental results are promising, highlighting the trade-off between performance and computational resources.

3. The analysis of the impact of visual tokens is valuable, providing meaningful conclusions that deepen our understanding.

**Weaknesses:**

1. Can you explain the choice of the MQT transformer design? Does it need to maintain sufficient capability for extracting information from varied tokens? For example, Q-Former has multiple layers to achieve effective information extraction.

2. The extraction process V=Q(Z,G) lacks correlation with textual information. In Fig 4, different text prompts should result in the same attention map due to this. Therefore, this design might be limited when dealing with complex images, where visual representations should maintain different semantics.

3. How does the model perform in text-rich scenarios, such as TextVQA? The sophisticated recognition ability might be significantly compromised in these cases.

4. Since the number of visual tokens affects performance, how does the model perform when the number of visual tokens exceeds the original 576 tokens?

5. It would be better to provide a comparison with the LLaVA baseline, using adaptive pooling or vanilla Q-former to adjust token numbers.

**Questions:**

As in LLaVA-1.5, the 576 visual tokens do not introduce significant computational overhead. It would be more meaningful to evaluate this method on a more extensive case like LLaVA-NEXT.

**Limitations:**

Please consider answer the questions above.

---

> ### Author Rebuttal · Authors · 2024-08-07
>
> Thank you for your review!
>
> 1. How many layers does MQT transformer need as compared to Q-Former?
>
> We studied the layers for MQT transformers at the early stage of our research. We didn’t find improved performance with multiple layers. This similar one layer design is also employed by Qwen-VL (Bai et al., 2023) which is a strong LVLM in the open-source community. Our extensive experiments in **Table 1** and comparison to Q-Former in Figure 3 (''With only 2 visual tokens, MQT-LLAVA outperforms InstructBLIP on all 8 tasks'') have demonstrated the sufficient capabilities of MQT design.
>
> 2.  Should text prompts be involved in the visual feature extraction process?
>
> Thanks for the detailed observation of our model. Most current LVLMs (except BLIP2 has an extra text encoder) do not use text prompts as input for extracting visual information. All the visual features are provided to LLM, together with the input question. LLM can then employ the textual information and identify useful visual features in answering the text question. We acknowledge this is a good suggestion in the lower number of visual tokens scenarios, whereas the model can pre-select the useful visual feature with textual information before extracting them. But an extra text understanding model needs to be properly designed for this purpose. We will leave this suggestion to future study of our work, but we believe this is orthogonal to the main claims of our paper.
>
> 3. TextVQA Performance
>
> Please refer to the general response’s Table 2 for added TextVQA results. We observe a very steady performance trend even when reducing the number of visual tokens to 16 tokens. In our paper’s Table 1, we have tested on 4 comprehensive evaluation benchmarks for Large Multimodal Models including MME,  MMMU, MMBench, and MM-Vet, where our model demonstrated robust performance on these tasks. All 4 benchmarks involving evaluation on OCR and text related tasks, we report the specific scores for these sub categories in our general response’s Table 3, please refer to it as well.  In summary, our model performs the best in these text related tasks with 3.2 points higher than LLaVA-1.5 on average.
>
> 4. How does the model perform when the number of visual tokens exceeds the original 576 tokens?
>
> Technically, the number of visual tokens is bounded by the original number since that’s the maximum visual information that a vision encoder can represent. But we did experiment with this interesting idea at an early stage. When we initialize the query tokens as sinusoidal embeddings similar to positional embeddings, then we can extrapolate the number of query tokens beyond the original limit. However, we didn’t find significant improvement in performance and the extrapolated visual tokens are also non intuitive in their meanings.
>
> 5. Adaptive Pooling or Vanilla Q-Former baseline.
>
> Our main idea is to train a model once and enable elastic inference of **any** number of visual tokens (under the maximum) for various computation resources and downstream tasks (line 7-9 and 12-14 in our Abstract). Adaptive pooling based approach will limit the choice of visual tokens to only several options. Whereas for Vanilla Q-Former, we conducted preliminary experiments at the starting point of our project, please refer to the detailed scores in our general response’s Table 2. In summary, vanilla Q-Former doesn’t demonstrate strong capabilities for our tasks.
>
> 6. Evaluation on more extensive heavy computation cases like LLaVA-Next.
>
> Our MQT design can be served as a plug-and-play module to other models (as mentioned by reviewer jmsV, too), we adopted LLaVA-1.5 as the base model to validate our methods and have demonstrated strong capabilities. We plan to extend our MQT method to videos and Interleaved image text tasks like LLaVA-Next as well. Recent work such as SlowFast-LLaVA (Xu et al., 2024)  from Apple have demonstrated promising results in this direction. However, these are out of the scope of this paper and can be considered as a future work direction. We have released source code and models for the community to try out the ideas for other models as well.

---

### Author Rebuttal · Authors · 2024-08-07

We sincerely thank all the reviewers for their time and their constructive reviews and suggestions. We are encouraged that the reviewers find that:

(1) Our Matryoshka Query transformer is interesting (Reviewer k14Y) and showing **incredible benefits and is a worthy contribution** (Reviewer  L7Jp). Our model handles the problem of arbitrary number of visual tokens which is **highly valuable** (Reviewer gY5g) and can be added as **plug-and-play** to any model (Reviewer jmsV).

(2) Our experiments are comprehensive (Reviewer jmsV). The experimental results are extensive (Reviewer L7Jp) and promising (Reviewer gY5g), highlighting the trade-off between performance and computational resources (Reviewer gY5g).

(3) The analysis of the impact of visual tokens is valuable (Reviewer gY5g). This good analysis goes beyond benchmark numbers (Reviewer L7Jp).

(4) Our paper is well-written and clear (Reviewers jmsV, k14Y, L7Jp) and presents a great job with interesting visualizations (Reviewers k14Y, L7Jp).

Thank all the reviewers for their help again! We believe the comments and revisions have made the paper stronger. Please find individual responses to your questions below.

---

### Decision · Program_Chairs · 2024-09-25

**Decision:**

Accept (poster)

**Comment:**

This paper unanimously receives positive rates thanks to the novel use of visual tokens and convincing experiments. Although overall reviews are positive, the rebuttal should be reflected in the final draft.